# Investigating Patients’ Preferences to Inform Drug Development Decisions: Novel Insights from a Discrete Choice Experiment in Migraine

**DOI:** 10.3390/ijerph18094916

**Published:** 2021-05-05

**Authors:** Aleksandra Torbica, Carla Rognoni, Rosanna Tarricone

**Affiliations:** 1CERGAS (Centre for Research on Health and Social Care Management), SDA Bocconi School of Management, Bocconi University, 20136 Milan, Italy; aleksandra.torbica@unibocconi.it (A.T.); rosanna.tarricone@unibocconi.it (R.T.); 2Department of Social and Political Sciences, Bocconi University, 20136 Milan, Italy

**Keywords:** migraine, discrete choice experiment, patients’ preferences, treatment characteristics

## Abstract

There is limited evidence on the scope and overall benefit of patient-centred drug development decisions. The present study assessed patients’ preferences for the characteristics of an ideal migraine treatment through a discrete choice experiment in order to inform decision-making and drug development processes. We investigated the preferences according to five treatment attributes identified from a systematic literature review and two focus group elicitations. The heterogeneity of preferences was also investigated. Overall, the respondents considered the presence of adverse events, duration of treatment effect, reduction of symptom intensity, speed of effect and cost born by the patient as the most relevant treatment features. As expected, the patients preferred treatments with lower levels of adverse events and costs and treatments with greater speed, duration of treatment effect and effectiveness in reducing symptom intensity. There was significant preference heterogeneity only for the presence of adverse events. Compared to men, women had significantly higher preferences for quicker treatment effect and limited adverse events and reported higher preferences for costly treatments. The results of our survey help address research and development strategies in the pharmaceutical industry and public policy regarding treatments that are clinically effective and responsive to the needs expressed by patients.

## 1. Introduction

Pharmaceutical companies are continuously investing in the research and development of products that are supposed to meet the needs of patients, physicians and payers while adhering to regulatory requirements. The development of a treatment to help improve patients’ lives should be rooted in a solid understanding of the challenges these patients face in their daily activities, their needs and the compromises they are willing to make to obtain relief. To ensure the creation of valuable treatments, all aspects of the healthcare system and treatment decisions need to be aligned with the needs of patients [1]. At present, there are limited data describing the scope and overall benefit of existing patient-centred drug development activities [2], and the assessment of patients’ preferences for treatment characteristics in this setting would increase the current evidence on the impact of diseases to support decision-making processes in the field of public policies and research and development in pharmaceutical industries.

Investigating patients’ preferences is essential to inform these decisions for all diseases, but it appears to be particularly important for chronic conditions that require patient commitment to their management, and this has important long-term consequences for patients’ personal and social lives. Chronic migraine may be considered a paradigmatic example in this respect.

Available data estimate that approximately 12% of adults worldwide develop some form of migraine throughout their lives [3,4]. Migraine is generally classified into episodic migraine (4 to 14 days of migraine a month) and chronic migraine (≥15 days of migraine per month). Women are most affected by migraine. In Italy, 6 million people suffer from migraines, and 4 million of the affected are women. Considering the plurality of costs associated with the management of the disease, migraine also has a significant socio-economic weight [5]. A study conducted in Italy estimated an annual cost for the management of a patient with migraine of EUR 4352, mainly composed by productivity losses (36%) and informal care (34%) [6]. Moreover, Italian women lose more days of work (16.8 per year versus 13.6 for males) and days of social life (26.4 per year versus 20) and are more likely to work in conditions of malaise (51.6 days per year versus 35.6). The study highlighted also the burden of this disease on caregivers, showing that 63% of patients received informal care from family members, on average for 4 days a month. Migraine has also a significant impact on patient’s quality of life, especially for women in relation to the interruption or limitation of daily activities [7]. Despite these evidences, a cultural gap persists on the migraine pathology that leads the disease to be underestimated, underdiagnosed and consequently undertreated or possibly erroneously managed [6].The literature includes different studies that evaluated patient preferences for migraine treatment characteristics. A few studies focused on specific treatments such as triptans [8,9], while other studies focused on preventive treatments only [10,11]. Most studies used ad hoc surveys to investigate patient preferences, but only two implemented discrete choice experiments [11,12], which allowed the assessment of the relative importance of different attributes. The studies in general were not limited to individuals with migraine but included patients with headache and did not consider gender differences in treatment preferences.

The available evidence highlights that both episodic and chronic migraine present a high social burden and represent an “unmet need” for public health that requires particular attention. Although there is increasing evidence supporting the role of gender in epidemiology and diagnosis, the identification of gender differences in migraine treatment and related efficacy is basically ignored [13]. In the presence of a disease that primarily affects women, gaps remain in gender-specific research at preclinical and clinical levels. The same gap occurs in the drug development process, which does not generally consider patient preferences and differences based on gender [14].

The current study expands the existing literature on patient preferences for an ideal treatment for migraine to overcome the identified gaps by considering the specific population of individuals with migraine, applying a solid methodology and investigating gender differences and heterogeneity in these preferences. We employed a discrete choice experiment (DCE) methodology to investigate the preferences of a heterogeneous group of patients for different characteristics of a hypothetical but realistic therapeutic regime for migraine. The ultimate aim is to improve the existing body of knowledge and provide an evidence base to inform patient-centred policies and the research and development decisions made by pharmaceutical industries.

## 2. Materials and Methods

To evaluate the preferences of individuals with migraine for treatments characteristics and evaluate how socio-economic characteristics, particularly gender, may influence those preferences, a DCE was implemented [15]. The DCE was performed to identify and estimate the relative importance assigned to different characteristics of a hypothetical treatment for migraine.

DCE is a technique to elicit preferences, which is based on the fact that goods or interventions may be described according to their characteristics (or attributes), and each attribute is represented by a defined number of dimensions, called attribute levels, on which the preferences of individuals for these goods depend [16]. For example, attributes for a migraine treatment may be route of administration (e.g., oral or intravenous), duration of the effect (e.g., 6 h, 8 h, or all day), speed of relief (e.g., 30 min, 1 h, or 2 h), etc.

A DCE consists of the presentation of two or more hypothetical scenarios, created by combining attribute levels, to participants, who choose among a number of alternative options [17]. Respondents trade-off between the attributes during the decision-making process and select the preferred option. The choices of the respondents indicate the preference or utility attached to a good or intervention and its attributes [18].

### 2.1. Development of DCE Attributes and Associated Levels

A systematic literature research was performed in March 2018 to retrieve the available evidence on the preferences of patients for the main features of migraine treatments. This preliminary activity allowed the identification of a possible list of treatment characteristics for further investigation via a discussion with a sample of individuals with migraine (see Appendix A for details).

The search query led to the selection of 14 studies that contained information about patient preferences [8,9,10,11,12,19,20,21,22,23,24,25,26,27], which were measured on the basis of more or less structured questionnaires to understand which characteristics of a treatment they considered most important. The population considered was mostly composed of patients with migraine, without distinction based on type (chronic or episodic). The treatment characteristics considered in the various published studies were resolution of symptoms (32%), efficacy (speed of effect, no recurrences, delay in the attack and persistence of effect) (30%), adverse events (14%), return to daily activities (11%), formulation (11%) and cost (3%). Although these studies investigated both genders, no differences in preferences emerged.

After collecting the available evidence in the international literature, we organized two focus groups to evaluate the characteristics of a hypothetical treatment that the patients considered most relevant in the national context. The focus group was a group interview led by a moderator who followed a structured outline and proposed stimuli to the participants [28]. The idea behind this method is that the considerations or evaluations expressed by each of the participants elicit reactions, comments and reflections from the others, which activates a spontaneous and intense discussion that leads to the sharing of further and more in-depth considerations. Each focus group lasted two hours and involved 8 adult patients who suffered from migraine with an average number of attacks per month greater than or equal to 4 (group 1: mean age 49 years, 37% males; group 2: mean age 45 years, 50% males). The focus group discussion started with the presentation of the participants with their personal characteristics of migraine and with the sharing of treatment features studied in the literature and derived by the literature search. After interactive discussion, each participant rated the treatment features from the most important to the less important. The following five most important characteristics were considered by the participants in the two focus groups:speed of effect (how quickly the treatment relieves the symptoms);reduction in the intensity of symptoms (efficacy-strength);duration of effect (efficacy-duration);adverse events;cost born by the patient.

Concerning the quantification of the levels of variation of these attributes, the participants reported that the effectiveness, in terms of speed in achieving the effect, should ideally be within 30 min of drug administration. Times from 30 to 90 min were acceptable, while times over 3 h were considered excessive. Regarding the efficacy of the treatment in terms of intensity of symptom reduction, acceptable levels were approximately 50%, with ideal values of symptom reduction of at least 90%. Patients believed that the effectiveness of the treatment should ideally last all day, or at least for 8 h, considering the workday. Furthermore, patients specified the possible side effects (i.e., tachycardia, drowsiness, daze, tingling, and gastrointestinal effects) and the preference for treatments with limited adverse events. Regarding the cost born by patients to purchase an ideal treatment, patients found a cost of EUR 50 per month acceptable, but they would be willing to bear a cost of up to EUR 200 per month to have resolution of all symptoms related to migraine. Table 1 reports the summary of attributes and levels discussed during the focus groups. Adverse events severity was classified according to FDA Adverse Event Reporting System (FAERS) [29].

### 2.2. Experimental Design

After the elicitation of attributes and associated levels, a DCE with three choice alternatives (i.e., two alternative choices and an opt-out option) was developed. Inclusion of an opt-out was deemed necessary because forcing respondents to make a choice on a treatment can lead to an overestimation of the utility for parameters [30]. The opt-out option, corresponding to “no treatment”, was associated with zero speed of effect, zero percent of symptoms reduction, no duration of effect, no presence of adverse events and no cost born by the patient.

The choice sets were combined using Ngene software [31]. From a full design (factorial), an optimal design (orthogonal in the differences—OOD) [32] was derived by considering the total number of choices between alternatives equal to 100 (D-optimality = 92.5%). The choice sets were grouped into five blocks. Each block comprised 20 choice sets, ordered differently across blocks. A sample size of minimum of 250 respondents was identified for parameter estimations for the DCE, and the blocks were distributed randomly among them [33].

### 2.3. Data Collection

A questionnaire representing the choice set was administered online in October–November 2018 to a sample of 466 adult patients suffering from migraine with an average number of attacks per month greater than or equal to 4 (see Appendix B). The web survey was implemented by an external company (Pepe Research https://www.peperesearch.it (accessed on 2 April 2021)), which selected respondents from its own database according to the characteristics described above. The database contains individuals that self-reported to be affected by one or more diseases. Individuals classified as migraineurs have been selected for the study. The first two questions of the survey served as a confirmation that the respondents were effectively affected by migraine.

The survey was pretested online on a sample of 27 respondents, who had the chance to report possible comprehension problems and difficulties in the questionnaire completion. This phase did not identify possible comprehension problems or difficulties in executing the tasks.

The following data categories were collected:Socio-demographic characteristics;Education level, professional status, and net annual income;Duration of the single migraine attack;Type of migraine (with or without aura);Average number of attacks per month.

The e-survey presented an explanation of the scope of the interview, a description of the different levels and attributes and 20 choices among the three alternatives. Each attribute was carefully explained using patient-friendly language. In order to simulate a clinically relevant decision context, we clarified in the questionnaire that patients had to choose only on the basis of the available given information and we explicitly specified an opt-out condition in line with good practice recommendations.

The participants’ task was to evaluate two alternatives (A and B) and the opt-out option (alternative C, no treatment) in each choice set and choose the option that reached the preference from their point of view. An example of a scenario with the choice of three alternatives is reported in Figure 1.

At the end of the survey, 5 questions investigated the importance of the considered attributes (see Table 1) through a Likert scale with 5 levels (not at all important, not very important, quite important, very important and extremely important). One question investigated the difficulties experienced during the choices among the proposed alternatives, and the possible responses were no difficulties, few difficulties, moderate difficulties, many difficulties and extreme difficulties.

### 2.4. Data Analysis

Respondent characteristics and responses were summarized using descriptive statistics. The DCE choices were first reviewed to determine whether respondents showed dominant preferences (i.e., frequently selected the set with the best level of a particular attribute) and in which cases the respondents chose the “opt-out” option. For each respondent, dominance scores were calculated for each attribute; a score of 1 was assigned for each question in case the respondent chose an attribute at its best level (e.g., maximum power of the effect or minimum cost). Therefore, a maximum score equal to 20 for an attribute was assigned to a respondent who chose the best level of this attribute for all 20 questions.

The DCE data were analysed within a random utility maximization framework [34,35], and a mixed logit regression analysis was used to construct a model of choice behaviour. In this framework standard deviations are generated to quantify preference heterogeneity for attributes and attribute levels [36].

The following model of choice behaviour (Model 1) was used to evaluate the impact of different attributes and levels on the preferences for alternative treatment approaches [37]:U_ijs_ = β_1_ (speed) + β_2_ (efficacy) + β_3_ (duration) + β_4_ (adverse events) + β_5_ (cost) + ε_ijs_
where U_ij_ is the utility individual i derives from choosing alternative j in choice situation *s*, β_1_–β_5_ are the coefficients of preference weights reflecting the desirability of the attributes and ε_ijs_ is the error term assumed independent and identically distributed (random component).

The output of a mixed logit model includes the means and standard deviations (SDs) of random coefficients and their respective confidence intervals (CIs). The mean coefficients represent the relative utility of each attribute conditional on other attributes, and the SDs reflect the degree of heterogeneity among the respondents (higher the value, higher the heterogeneity).

The inclusion of the cost born by the patient among the attributes allows the estimation of the willingness to pay (WTP) [38]. WTP is interpreted as an estimate of the relative values assigned to an attribute included in the choice set, and it is expressed in monetary terms. A WTP analysis was performed starting from the results of the regression analyses for Model 1 and males and females separately. We calculated the WTP for nonmonetary attributes as the ratio of the cost coefficient and mean coefficients for the attributes.

We extended the baseline model to examine the drivers of response heterogeneity. More specifically, to investigate the extent to which preferences were driven by respondents’ characteristics, Model 1 was extended to Model 2, which considered interactions between attributes and patients’ gender and age. For this model, continuous (i.e., age) and categorical (gender) interaction variables were considered.

The random parameters for all attribute levels were estimated assuming a normal distribution. The Akaike information criterion (AIC) was applied to compare the goodness of fit and to test the extended model against the model with no interactions.

Data analyses were undertaken using Stata (StataCorp. 2019. Stata Statistical Software: Release 15. College Station, TX: StataCorp LP).

## 3. Results

All of the 466 participants who enrolled completed the e-survey. The participants had an average age of 43 years (range 18–77), and 66% were women. A statistically significant difference emerged for age between the two genders, with males being older than females (45 years vs. 41 years). Working respondents reported that they had a full-time job in most cases (72%). The annual income ranges were equally distributed over the total number of respondents. Only 7% of respondents preferred to not declare their income. Statistically significant differences emerged between the genders for declared income, with lower incomes for females, who also reported less remunerative working roles.

Considering the characteristics of migraine, females showed a greater and significant duration of attack than males; a duration of at least 4 h was reported by 70% of females (= 45% duration 4–24 h + 25% duration 2–3 days) and by 49% of males (=41% duration 4–24 h + 8% duration 2–3 days). Aura affected more females than males (45% vs. 35%).

The socio-demographic characteristics of the sample, including disease characteristics, are summarized in Table 2.

Regarding DCE, the participants reported 9320 responses (20 questions each), with a significantly high number of cases (38%) in which respondents chose the opt-out option. In addition, 25% of respondents reported having encountered moderate-to-extreme difficulty in choosing among the alternatives.

The analysis of dominant preferences revealed that only eight respondents expressed dominant preferences (score = 20) for the presence of adverse events, seven for speed of effect and seven for cost. The dominant preferences of other attributes were limited. The overall pattern of results suggests that the presence of side effects, speed of effect and the cost born by the patients were the most important factors for respondents in deciding which treatment they would select.

The full results of mixed logit models are presented in Table 3.

In Model 1, all the attributes significantly impacted the probability of choosing an alternative (*p*-values < 0.05). The negative sign of the coefficient for speed of effect (−0.00328) indicates that as the time to obtain a treatment effect increased, the patients’ likelihood of choosing this scenario decreased. The same results were found for adverse events and cost. In contrast, respondents preferred higher levels for strength of efficacy and duration of effect. Only the presence of adverse events showed a significant preference heterogeneity (standard deviation *p*-value = 0.003). These results are consistent with the indications received during the focus groups.

In Model 2, all the attributes (β_1_–β_5_) significantly impacted the probability of choosing an alternative. The presence of adverse events maintained heterogeneity in the preferences (standard deviation *p*-value = 0.007).

Gender interactions were statistically significant for speed of effect, presence of adverse events and cost. The results revealed that, compared to men, women had significantly higher preferences for quicker treatment effect and the presence of limited adverse events, and they reported higher preferences for more costly treatments. Age did not seem to influence the preferences of patients.

According to the Akaike information criterion, Model 2 showed a better fit than Model 1.

The WTP analysis performed on the base-case mixed logit model (Model 1) showed that respondents would be willing to pay EUR 0.61 to anticipate the effect of the treatment by one minute, everything else being equal. Patients would be willing to pay EUR 4.52, €16.66 and €141.68 to acquire a one percentage point increase in the strength of symptom reduction, for having an additional hour of effect duration and for de-escalating the adverse events, respectively, with everything else being equal. These results show that the reduction of adverse events is the most important dimension for which patients would be willing to pay the highest amount.

The WTP analysis performed separately on males and females showed a higher willingness to pay for women compared with men for all of the considered attributes (EUR 0.98 vs. EUR 0.18 to anticipate the effect of the treatment by one minute, EUR 5.25 vs. EUR 3.72 to gain a one percentage point increase in the strength of symptom reduction, EUR 19.60 vs. EUR 12.62 for having an additional hour of treatment effect duration and EUR 172.69 vs. EUR 97.10 for de-escalating the adverse events).

The responses on the Likert scale were consistent with the results obtained in the DCE, and they confirmed the presence of adverse events as the most important treatment feature. The following other attributes, in order of importance, were efficacy-strength, speed of effect, duration of the effect and cost born by the patient (see Table 4).

## 4. Discussion

For many patients suffering from migraine, finding the right combination of clinical treatment and routine is a lifelong challenge. Where different treatment strategies are available, it is of the utmost importance to support the treatment choices of patients to improve their compliance with therapies.

The present study employed a DCE to investigate gender differences in preferences for a set of attributes of treatment for migraine. The key feature of the study model was the use of a mixed logit model that allowed us to account for heterogeneity in preferences driven by observable characteristics (primarily gender) and test for the existence of residual significant heterogeneity in nonobservable characteristics.

The presence of adverse events, duration of the treatment effect, reduction of the intensity of the symptoms, speed of the effect and cost born by the patient were, in that order, the attributes considered most relevant by the respondents. These data are consistent with our international literature review, which reported these five attributes as the most important in 79% of studies. These results are consistent with another DCE involving 510 patients with migraine in the USA [12], which investigated the severity and duration of symptoms in headache and postheadache phases, the limitation in activity and the chance of migraine attack recurrence. The study showed that hypothetical treatments that relieved and shortened symptoms during the postheadache phase offered significant benefits to individuals with migraine. The same results were found in another DCE study [11] that focused on migraine prevention, in which 72% of respondents rated treatment effectiveness as the most important aspect. The present study revealed that all the attributes significantly impacted the probability of choosing an alternative and that, in general, respondents preferred lower levels for speed of effect (quick response), adverse events and cost and higher levels for strength of efficacy and duration of effect.

Only the presence of adverse events was associated with significant heterogeneity with respondents’ preferences. The interaction analysis also highlighted that compared to men, women had significantly higher preferences for quick treatment effect and the presence of limited adverse events and that women reported higher preferences for costly treatments. Women, who in general report a worse quality of life and worse symptoms than men, seemed willing to pay more than males to receive a more effective treatment. In fact, 45% of female migraine patients have aura (a set of sensory, motor or verbal disturbances) compared with 35% of male migraine patients, with a statistically significant difference between the two groups. Based on this, it is reasonable for women to be willing to pay more for migraine treatment, irrespective of preference for treatment methods, in order to alleviate these heavy symptoms. Age did not influence patients’ preferences.

As direct beneficiaries of health services, patients have a widespread awareness of the impact and the effects of a treatment on their condition and on different aspects of their life. The incorporation of patient preferences in the drug development process may be of paramount value. Different initiatives have started to integrate the patient’s voice into therapeutic development and regulatory review. An example is the Patient-Focused Drug Development (PFDD) initiative [39], which includes meetings that aim to help the Food and Drug Administration (FDA) understand the burden of disease from the patient perspective and to gain an appreciation for the factors that are taken into account by patients when a treatment is chosen. Similar developments started also in Europe, with the incorporation of patient preferences into the assessment of oncology treatments by the European Medicines Agency (EMA) [40]. Again, the European Patients Academy on Therapeutic Innovation (EUPATI) [41] focused on education and training to increase the capacity and capability of patients to be valuable contributors to medicines research and development. The PREFER initiative, built upon the experiences and outcomes of previous projects and initiatives, aims at establishing recommendations to support development of guidelines for industry, regulatory authorities and HTA bodies on how and when to include patient perspectives on benefits and risks of medicinal products [42].

This study provides novel insight into this growing body of literature, but some limitations need to be recognized. First, involved patients were recruited via an online survey, and this population may be biased towards the ability to use a computer or a mobile device; moreover, only respondents with internet access were able to complete the questionnaire [43]. Another limitation relates to the selection of migraineurs; although they were selected from a pre-existing database managed by a market research company (Pepe Research), no official documentation on migraine diagnosis was requested in order to include an individual in the database and respondents may either have identified themselves as being affected by migraine or may have received an official diagnosis by a physician. This could have increased the heterogeneity in the sample analysed.

Third, in the range of levels presented, a significant portion of the sample did not appear to use a trade-off between the attributes and chose the opt-out option in 38% of cases. Fourth, as a cross-sectional hypothetical experiment among respondents, the elicited preferences described in this study may change over time, especially once respondents experience different treatment strategies for migraine. Therefore, while the results are internally valid, generalizability beyond the study context cannot be explicitly guaranteed [44]. However, these concerns are not peculiar to our own study but represent general concerns pertaining to the application of stated preferences techniques [44].

## 5. Conclusions

To date, there is no resolutive treatment for migraine. Nausea, visual disturbances and hypersensitivity to sounds, smells and light make the disorder more complex, and patients may not even be able to get out of bed on the worst days. Although the range of pharmacological opportunities in use today is being enriched, the involvement of patients in drug development activities and in the collection of their preferences seems not routinely used.

Our study showed that patients prefer treatments with lower levels of adverse events and costs and treatments with greater speed, duration of treatment effect and effectiveness in reducing symptom intensity. Female migraine patients are more inclined to choose faster treatments, fewer side effects, but also more expensive treatment options. The results of our survey can help address the research and development strategies of the pharmaceutical industry towards treatments that are clinically effective and responsive to the needs expressed by the patients.

## Figures and Tables

**Figure 1 ijerph-18-04916-f001:**
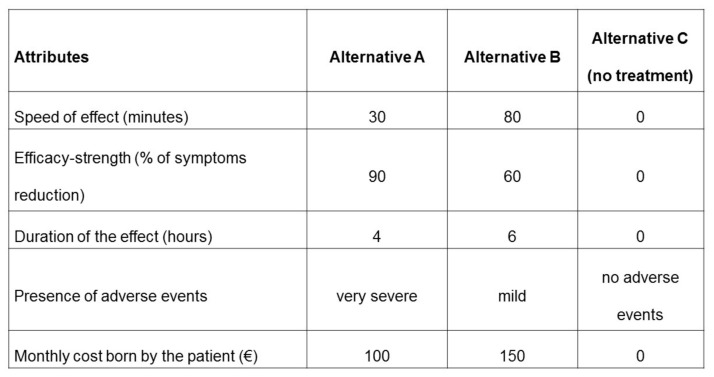
Example of choice set.

**Table 1 ijerph-18-04916-t001:** Description of attributes and levels.

Attribute/Levels	Level 1	Level 2	Level 3	Level 4
Speed of effect (minutes)	30	80	130	180
Efficacy-strength	60%	70%	80%	90%
Efficacy-duration (hours)	4	6	8	10
Adverse events	mild	moderate	severe	very severe
Monthly cost born by the patient	EUR 50	EUR 100	EUR 150	EUR 200

**Table 2 ijerph-18-04916-t002:** Socio-demographic characteristics of the sample of respondents with the characteristics of migraine.

Parameter	Total Population	Males	Females	*p*-Value
Gender	466	159 (34%)	307 (66%)	
Mean age (years)	43 (18–77)	45	41	<0.0001
Education level				
Primary school	8%	6%	9%	0.62
High school diploma	52%	54%	51%	
Bachelor’s degree	14%	12%	15%	
Master’s degree	21%	22%	20%	
Doctorate	6%	7%	5%	
Professional activity				
White collar	62.00%	78.60%	53.50%	<0.0001
Blue collar	8.80%	12.60%	6.80%	
Retiree	2.60%	2.50%	2.60%	
Homemaker	13.50%	1.90%	19.50%	
Student	5.40%	0.60%	7.80%	
Unemployed	7.70%	3.80%	9.80%	
Income ranges declared by the workers (annual net)				
Less than EUR 15,000	21%	10%	29%	<0.0001
EUR 15,000–19,999	21%	14%	26%	
EUR 20,000–29,999	30%	36%	26%	
EUR 30,000 or more	28%	40%	19%	
Duration of migraine attack, frequency and symptoms				
Few minutes	2%	4%	2%	<0.0001
Up to 3 h	35%	47%	28%	
From 4 to 24 h	43%	41%	45%	
2–3 days	20%	8%	25%	
Average number of attacks per month	7.2	7.5	7.0	0.85
Number of attacks per month from 4 to 8	76%	72%	78%	0.339
Number of attacks per month from 9 to 15	19%	23%	18%	
Number of attacks per month higher than 15	5%	6%	5%	
Presence of aura	42%	35%	45%	0.037

**Table 3 ijerph-18-04916-t003:** Results of the mixed logit Models 1 and 2.

Attributes	Model 1				Model 2			
	Mean Coefficient Values		Standard Deviations		Mean Coefficient Values		Standard Deviations	
	β	*p*-Value	β	*p*-Value	β	*p*-Value	β	*p*-Value
Speed of effect (minutes)	−0.0033	<0.00001	0.0005	0.8500	−0.0018	0.1830	0.0017	0.4440
Efficacy-strength (% of symptoms reduction)	0.0242	<0.00001	0.0002	0.9030	0.0244	<0.00001	0.0002	0.9040
Duration of the effect (hours)	0.0891	<0.00001	−0.0008	0.9680	0.0903	0.0070	−0.0003	0.9880
Presence of adverse events	−0.7580	<0.00001	−0.2012	0.0030	−0.4484	<0.00001	−0.2404	<0.00001
Monthly cost born by the patient (€)	−0.0054	<0.00001	0.0009	0.5220	−0.0046	0.0010	0.0020	0.1670
Female*speed of effect					−0.0037	<0.00001		
Female*efficacy-strength					0.0037	0.0670		
Female*duration of the effect (hours)					0.0237	0.1280		
Female*presence of adverse events					−0.3346	<0.00001		
Female*monthly cost born by the patient					0.0012	0.0690		
Age*speed of effect					0.0000	0.5960		
Age*efficacy-strength					0.0000	0.7310		
Age*duration of the effect (hours)					−0.0003	0.6850		
Age*presence of adverse events					−0.0031	0.0660		
Age*monthly cost born by the patient					0.0000	0.1600		
N. observations	27,681				27,681			
N. of respondents	466				466			
Log-likelihood	−8664				−8550			
Akaike information criterion (AIC)	17,338				17,114			

**Table 4 ijerph-18-04916-t004:** Assessment of the importance of the considered attributes on the Likert scale.

Grading	Speed of Effect	Efficacy-Strength	Duration of the Effect	Presence of Adverse Events	Monthly Cost
Extremely important	33.9%	30.0%	26.6%	58.2%	26.0%
Very important	33.0%	40.6%	37.8%	22.3%	30.5%
Quite important	28.5%	27.5%	31.1%	14.6%	32.6%
Not very important	4.1%	1.7%	3.9%	4.3%	8.6%
Not at all important	0.4%	0.2%	0.6%	0.6%	2.4%

## Data Availability

The data presented in this study are available on request from the corresponding author. Only aggregated data can be shared due to ethical restrictions.

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
