# Peer review of "Investigating Patients’ Preferences to Inform Drug Development Decisions: Novel Insights from a Discrete Choice Experiment in Migraine"

_ijerph, 2021, doi:10.3390/ijerph18094916_

Round 1

Reviewer 1 Report

In this investigation, the authors observed the preferences of migraine patients in making treatment decisions and they analyzed five important characteristics of patients with migraine during therapy, including therapeutic effect of speed, treatment cost, side effects, symptom reduction, and duration of effectiveness. The authors found that female migraine patients are more inclined to treat faster, choose fewer side effects, but also more expensive treatment options. This study is useful for clinical treatment of migraine. However, the authors need to address the following concerns before being ready for publication.

  1. Table 2 shows that 45% of female migraine patients have aura compared with 35% of male migraine patients, a significant difference between the two. Based on this, it is reasonable for women to pay more for migraine treatment, which may have nothing to do with their preference for treatment methods, but with the actual cost of treatment. The authors need to explain this result in the discussion.
  2. The authors mentioned "Considering the characteristics of migraine, females showed a greater and significant duration of attack than males (duration greater than or equal to 4 hours: 70% vs. 49%)", which is not consistent with the statistical results shown in Table 2. Please explain.
  3. The conclusions described in the abstract are inconsistent with those in the manuscript.

Author Response

We would like to thank the Reviewer for the comments and suggestions. Please find below the required changes (marked with R:) in response to the remarks for your consideration. We have carefully revised the manuscript and we hope that the changes made improved the quality of the manuscript and made it suitable for publication.

Best Regards,

The authors

1. Table 2 shows that 45% of female migraine patients have aura compared with 35% of male migraine patients, a significant difference between the two. Based on this, it is reasonable for women to pay more for migraine treatment, which may have nothing to do with their preference for treatment methods, but with the actual cost of treatment. The authors need to explain this result in the discussion.

R: We thank the Reviewer for the suggestion. We included a discussion on this point:

“Women, who in general report a worse quality of life and worse symptoms than men, seemed willing to pay more than males to receive a more effective treatment. In fact, 45% of female migraine patients have aura (a set of sensory, motor or verbal disturbances) compared with 35% of male migraine patients, with a statistically significant difference between the two groups. Based on this, it is reasonable for women to willing to pay more for migraine treatment, irrespective of preference for treatment methods, in order to alleviate these heavy symptoms.”

2. The authors mentioned "Considering the characteristics of migraine, females showed a greater and significant duration of attack than males (duration greater than or equal to 4 hours: 70% vs. 49%)", which is not consistent with the statistical results shown in Table 2. Please explain.

R: The migraine attack duration has been discretized according to different time ranges:

A) Few minutes

B) Up to 3 hours

C) From 4 to 24 hours

D) 2-3 days

Longer durations (at least 4 hours) have been reported for females (the sum of the % for C and D was 45%+25%=70%) compared with males (for whom the sum of the % for C and D was 41%+8%=49%).

We thank you for highlighting this unclear point. We included a more detailed description of the calculation in the text:

“a duration of at least 4 hours was reported by 70% of females (= 45% duration 4-24 hours + 25% duration 2-3 days) and by 49% of males (= 41% duration 4-24 hours + 8% duration 2-3 days).”

3) The conclusions described in the abstract are inconsistent with those in the manuscript.

R: We thank the Reviewer for highlighting this discrepancy. We modified and completed the Conclusions:

“Our study showed that patients prefer treatments with lower levels of adverse events and costs and treatments with greater speed, duration of treatment effect and effectiveness in reducing symptom intensity. Female migraine patients are more inclined to choose faster treatments, fewer side effects, but also more expensive treatment options. The results of our survey can help address the research and development strategies of the pharmaceutical industry towards treatments that are clinically effective and responsive to the needs expressed by the patients.”

Reviewer 2 Report

This is a well-done study and a well-written manuscript. I enjoyed reading it.

  1. Regarding methods- were all of the survey participants diagnosed with migraine by a physician? And if so, what kind of physician? Primary care physician? Neurologist? Headache specialist? You indicate that the survey was administered to migraine patients, but it is unclear whether these patients self-identified as having migraines, or had an official diagnosis by a physician. Please clarify. If they were not diagnosed by a physician, this should be listed as a limitation in the discussion section. 
  2. I think that lines 388-390 in the conclusion are worded too strongly. While there is definitely room to improve, I don't think the data supports that all drug development activities are not patient-centered and don't take patient preferences into account. I suggest rewording that.

Author Response

We would like to thank the Reviewer for the comments and suggestions. Please find below the required changes (marked with R:) in response to the remarks for your consideration. We have carefully revised the manuscript and we hope that the changes made improved the quality of the manuscript and made it suitable for publication.

Best Regards,

The authors

1. Regarding methods- were all of the survey participants diagnosed with migraine by a physician? And if so, what kind of physician? Primary care physician? Neurologist? Headache specialist? You indicate that the survey was administered to migraine patients, but it is unclear whether these patients self-identified as having migraines, or had an official diagnosis by a physician. Please clarify. If they were not diagnosed by a physician, this should be listed as a limitation in the discussion section. 

R: We thank the Reviewer for pointing out this lack. In the text we specified that

“The database contains individuals that self-reported to be affected by one or more diseases. Individuals classified as migraineurs have been selected for the study. The first two questions of the survey served as a confirmation that the respondents were effectively affected by migraine.”

We also discussed this point as a study limitation in the paper discussion:

“Another limitation relates to the selection of migraineurs; although they were selected from a pre-existing database managed by a market research company (Pepe Research), no official documentation on migraine diagnosis was requested in order to include an individual in the database and respondents may either have identified themselves as being affected by migraine or may have received an official diagnosis by a physician. This could have increased the heterogeneity in the sample analysed”.

2. I think that lines 388-390 in the conclusion are worded too strongly. While there is definitely room to improve, I don't think the data supports that all drug development activities are not patient-centered and don't take patient preferences into account. I suggest rewording that.

R: Thanks for the suggestion. We reworded the sentence into:

“Although the range of pharmacological opportunities in use today is being enriched, the involvement of patients in drug development activities and in the collection of their preferences seems not routinely used.”

Reviewer 3 Report

Interesting can be improved in adding elements if possible regarding age of participants. Sentences can be summarize for easy reading and comprehension of the text. Explanation of WHY adding socio cultural element should be provide, is there any correlation with migraina. Is there specific impact of socio cultural with migraina, ref?Please try to clarify and summarize on main elements.

Author Response

We would like to thank the Reviewer for the comments and suggestions. Please find below the required changes (marked with R:) in response to the remarks for your consideration. We have carefully revised the manuscript and we hope that the changes made improved the quality of the manuscript and made it suitable for publication.

Best Regards,

The authors

Interesting can be improved in adding elements if possible regarding age of participants.

R: The mean age of the respondents was 43 years (range 18-77) as we reported in Table 1. The age parameter has been included in Model 2 in order to investigate the interactions between treatment characteristics (attributes) and respondents’ characteristics (age and gender). The analyses showed that age did not influence the preferences of patients.

Sentences can be summarize for easy reading and comprehension of the text.

R: We thank the Reviewer for the suggestion. In fact the Methods section was quite complex. We tried to simplify the section by removing few technical aspects and by including few examples in order to improve understandability.

Explanation of WHY adding socio cultural element should be provide, is there any correlation with migraina. Is there specific impact of socio cultural with migraina, ref?

R: We included a description of the socio cultural impact of migraine with related references:

“A study conducted in Italy estimated an annual cost for the management of a patient with migraine of 4,352€, mainly composed by productivity losses (36%) and informal care (34%) [6]. Moreover, Italian women lose more days of work (16.8 per year versus 13.6 for males) and days of social life (26.4 per year versus 20) and are more likely to work in conditions of malaise (51.6 days per year versus 35.6). The study highlighted also the burden of this disease on caregivers, showing that 63% of patients received informal care from family members, on average for 4 days a month. Migraine has also a significant impact on patient's quality of life, especially for women in relation to the interruption or limitation of daily activities [7]. Despite these evidences, a cultural gap persists on the migraine pathology that leads the disease to be underestimated, underdiagnosed and consequently undertreated or possibly erroneously managed [6].”

Please try to clarify and summarize on main elements.

R: As stated above, we tried to simplify the description of the technical aspects in the Methods section.

Round 2

Reviewer 1 Report

The authors have addressed my concerns. I have no further questions.